# rcprd: An R package to simplify the extraction and processing of Clinical Practice Research Datalink (CPRD) data, and create analysis-ready datasets

Alexander Pate ®*, Rosa Parisi, Evangelos Kontopantelis, Matthew Sperrin

Division of Informatics, Imaging and Data Sciences, University of Manchester

* alexander.pate@manchester.ac.uk

## Abstract

The Clinical Practice Research Datalink (CPRD) is a large and widely used resource of electronic health records from the UK, linking primary care data to hospital data, death registration data, cancer registry data, deprivation data and mental health services data. Extraction and management of CPRD data is a computationally demanding process and requires a significant amount of work, in particular when using R. The **rcprd** package simplifies the process of extracting and processing CPRD data in order to build datasets ready for statistical analysis. Raw CPRD data is provided in thousands of.txt files, making querying this data cumbersome and inefficient. **rcprd** saves the relevant information into an SQLite database stored on the hard drive which can then be queried efficiently to extract required information about individuals. **rcprd** follows a four-stage process: 1) Definition of a cohort, 2) Read in medical/prescription data and save into an SQLite database, 3) Query this SQLite database for specific codes and tests to create variables for each individual in the cohort, 4) Combine extracted variables into a dataset ready for statistical analysis. Functions are available to extract common variable types (e.g., history of a condition, or time until an event occurs, relative to an index date), and more general functions for database queries, allowing users to define their own variables for extraction. The entire process can be done from within R, with no knowledge of SQL required. This manuscript showcases the functionality of **rcprd** by running through an example using simulated CPRD Aurum data. **rcprd** will reduce the duplication of time and effort among those using CPRD data for research, allowing more time to be focused on other aspects of research projects.

**Data availability statement:** The package developed in this study is available from the Comprehensive R Archive Network at https://CRAN.R-project.org/package=rcprd, or can be installed from the following GitHub repository: https://github.com/alexpate30/rcprd. The data for this worked example, which is simulated completely at random, is provided within the package. For the case-study we used synthetic patient-level data from the CPRD that are not publicly available. However, researchers can access these databases without a fee by contacting CPRD. Details of the application process and conditions of access are available at: https://www.cprd.com/synthetic-data and https://www.cprd.com/data-access. Code for implementing the case study is provided in the supplementary material, and is available at the following GitHub page: https://github.com/manchester-predictive-healthcare-group/CHI-CHARIOT/. The results in this paper were obtained using R version 4.4.0 with packages dplyr, tidyr, data.table, fastmatch, RSQLite, stringr, and all packages used are available from the Comprehensive Archive Network (CRAN).

**Funding:** This research was funded by The National Institute for Health Research (NIHR) School for Primary Care Research (SPCR) (reference: NIHR SPCR-2021-2026, grant number 648) and Endeavour Health Charitable Trust. The views expressed are those of the authors and not necessarily those of the NIHR, the Department of Health and Social Care, or Endeavour Health.

**Competing interests:** The authors have declared that no competing interests exist.

# 1 Introduction

The Clinical Practice Research Datalink (CPRD) is a large resource of Electronic Health Records from the UK, owned by the UK Medicines and Healthcare products Regulatory Agency (MHRA), and containing information on demography, medical history, test results and drug use of individuals registered with a general practice. The MHRA maintain two databases, CPRD GOLD, [1] which contains data from general practices using the Vision computer system, and CPRD Aurum [2], which contains data from general practices using the Egton Medical Information Systems (EMIS) computer system, EMIS Web. The primary care data is linked to hospital data, death registration data, cancer registry data, deprivation data and mental health services data, enabled by NHS digital [3]. As of 2016, the EMIS Web computer system was used by 4199 (56%) of the general practices in England [4]. As of September 2024, CPRD Aurum contained data on 47 million (16 million currently registered) individuals from 1,784 (1,596 currently contributing) general practices across the United Kingdom [5], and CPRD GOLD contained data on 21 million (2.9 million currently registered) individuals [6]. CPRD is a widely used resource: since 2019, a PubMed search reveals there have been 540 studies published which contain "CPRD" in the title or abstract. Extraction of CPRD data and transformation into a format ready for statistical analysis is computationally demanding and requires a significant amount of work. There is limited published software available to aid researchers in the extraction and processing of CPRD data [9].

**rEHR** [7] is an R package for manipulating and analysing electronic health record data, which works by creating an SQLite database on a fixed storage device (i.e., a disk drive), which is then subsequently queried to extract relevant information, faster than with conventional statistical analysis software. **rEHR** was designed to be database agnostic, and contains functionality for longitudinal data extraction, cutting data by time-varying covariates, matching controls to cases, converting the units of test data, and creating clinical code lists. **rEHR** is however no longer maintained, it works with an older version of R (3.3.2) and has been archived on the Comprehensive R Archive Network (CRAN). The **aurumpipeline** package [8] contains functions to clean and process CPRD Aurum data, which works by storing the data as parquet files on the disk drive, which are then subsequently queried to extract relevant data. However, **aurumpipeline** is not available on CRAN and is not provided with any reproducible examples. The R package **drugprepr** [9] implements the algorithm of [10] for preparing drug exposure data extracted from CPRD, however it does not deal with the initial data extraction and storing of data.

Given the many studies using CPRD data, and the limited availability software for data processing, this indicates that a large amount of research time is being spent duplicating the work of others in order to extract CPRD data. This study introduces **rcprd**, an R package designed to assist researchers in working with CPRD data, and creating datasets which are 'analysis-ready'. The main problem when working with CPRD data is the size of the raw data. Data on over 47 million individuals results in thousands of raw.txt files, and Terabytes of data, which can be cumbersome to work

with. This is a particular issue for R users, as its infeasible to read all this data into the R workspace simultaneously, as R operates using physical memory (RAM). As suggested by [7], **rcprd** bypassess this problem by creating an SQLite database which can then be queried for data of interest in order to build an analysis-ready dataset. **rcprd** then simplifies the process of querying the SQLite database with functions to extract variables such as "most recent test result", "time until first event", or "history of a specified condition". The SQLite database must be created on a secure device or server which aligns with the data storage requirements of CPRD.

We start by discussing the structure of CPRD Aurum data and the approach taken by **rcprd** for processing this data, which draws heavily on the work of [7]. We then run through a worked example to showcase the functionality of **rcprd**, which has two main groups of functions. The first are to extract and store the data in a consistent manner. The second group is to query this data to extract patient level variables. **rcprd** currently focuses on CPRD Aurum, as opposed to CPRD GOLD, given there has been a considerable drop in the number of practices utilising Vision software in the last 10 years, limiting the research utility of the CPRD GOLD database. However, **rcprd** has been designed to work with both CPRD Aurum and GOLD data, and functionality to formally accommodate CPRD GOLD data will be added in the near future.

## 2 Data structure and extraction process

### 2.1 Structure of CPRD aurum data

We first define the terminology which will be used throughout this article:

- *Raw data*: The raw data provided to the user by CPRD.

- *Cohort*: A cohort of individuals that meet the inclusion/exclusion criteria for a given research question. In this setting, the cohort is ultimately a vector of patient id's.

- *Analysis-ready dataset*: A data frame to which statistical models can be fitted, with one row for each individual in the cohort, and a column for each variable of interest, for example, age at cohort entry, or most recent BMI score prior to cohort entry. For longitudinal analyses, such data frames can be concatenated, with a variable indicating the time point at which the data was extracted.

The raw CPRD Aurum data is split into eight different file types: Consultation, DrugIssue, Observation, Patient, Practice, Problem, Referral, Staff. The data specification is available here: [11]. For most research questions, the relevant files are Patient, Observation and DrugIssue. The Patient file contains information about registration into the database, date of death or lost to follow up, year of birth and gender. This file will be required to define a cohort. The observation file contains all medical diagnoses and tests, while DrugIssue contains information on prescriptions. Medical observations are identified by their *medcodeid*, whereas prescriptions are identified through their *prodcodeid*.

In order to facilitate data transfer, this data is commonly split by CPRD into numerous smaller files. The different patient files are denoted by the string *set1*, *set2*, *set3* in the file name. Individuals in the same patient file will have the corresponding string (*setX*) in the files containing their medical or prescription data. However, there will be more than one Observation and DrugIssue file corresponding to each patient file. For example, the observation files for patients in *set1*, will have *set1* in their file name, and then an extra suffix 1, 2, 3, etc. The same is true for the DrugIssue files. The naming structure for these is as follows:

- aurum_allpatid_set*X*_extract_patient_001.txt

- aurum_allpatid_set*X*_extract_observation_0*Y*.txt

- aurum_allpatid_set*X*_extract_drugissue_0*Y*.txt

where $X \in \{1, 2, 3, ...\}$ and $Y \in \{01, 02, 03, ...\}$. Note that the prefix to the file names may vary (i.e., the 'aurum_allpatid' part) however we expect the naming convention with regards to 'set$X$', file type, and '0$Y$' to remain consistent. If this changes in the future, we will endeavour to update the **rcprd** as soon as possible.

## 2.2 Recommended process for extraction

Our recommended process for developing an analysis-ready dataset is detailed in Box 1.

---

### Box 1:–Flow diagram for recommended data extraction process

Step 1: Extract initial cohort using extract_cohort() and apply initial inclusion/exclusion criteria which can be applied using only the patient file.

Q1: Do the inclusion/exclusion criteria depend on primary care data?

- If yes -> (proceed to step 2)

- If no -> (proceed to Q2)

Step 2: For patients that meet inclusion/exclusion criteria from step 1, add relevant primary care data into an SQLite database using cprd_extract(). See section 3.2.

Step 3: Query SQLite database in order to apply further inclusion/exclusion criteria. See section 3.3.

Q2: Do the inclusion/exclusion criteria depend on linked data?

- If yes -> (proceed to step 4)

- If no -> (proceed to step 5)

Step 4: Request type 1 linked data* for individuals and apply remaining inclusion/exclusion criteria.

Step 5: For patients in the final cohort, add relevant primary care data into an SQLite database using cprd_extract(). If an SQLite database was already created in step 2, and only a small number of individuals were excluded in steps 3 and 4, consider skipping this step. However, if a large number of individuals were excluded, it is worthwhile to create the new SQLite database, as it will be much smaller and future queries for extracting variables will run quicker. See section 3.2.

Step 6: Query this SQLite database for specific codes and tests to create variables for each individual in the cohort. These are stored as.rds objects, which R data analysts will be familiar with. See section 3.3.

Step 7: Combine extracted variables into an analysis-ready dataset, also stored as an.rds object.

*type 1 linked data is defined as "linked data required in order to finalise the study population"

---

This process aligns with the process implemented by CPRD when cohort inclusion/exclusion criteria are dependent on linked data (see Q2 and step 4).

---

**Table 1. Table of rcprd functions.**

| Function | Use |
|---|---|
| **Step 1: Defining a cohort** | |
| *extract_cohort* | Create a cohort based on the raw Aurum patient files |
| **Step 2: Reading in data and creating an SQLite database** | |
| *connect_database* | Open a connection with a SQLite database |
| *add_to_database* | Add an individual raw Aurum data.txt files to SQLite database |
| *cprd_extract* | Add all raw Aurum data.txt files of a certain type to SQLite database |
| **Step 3: Querying the SQLite database to extract variables (extracting common variable types)** | |
| *extract_ho* | Extract a binary variable whether individual has a specified code recorded prior to index date, e.g., a 'history of' type variable. |
| *extract_time_until* | Extracts a time-to-event/survival variable: time from index date until event of interest occurs or individual is censored. |
| *extract_test_data* | Extracts most recent valid test result(s) in a given time period relative to the index date. |
| *extract_test_data_var* | Extracts the standard deviation of all valid test results in a given time period relative to the index date. |
| **Step 3: Querying the SQLite database to extract variables (extracting specific variables)** | |
| *extract_bmi* | Extracts BMI |
| *extract_cholhdl_ratio* | Extracts cholesterol/high density lipoprotein ratio |
| *extract_diabetes* | Extracts diabetes status (Absent, type 1, type 2) |
| *extract_smoking* | Extracts smoking |
| **Step 3: Querying the SQLite database to extract variables (database queries and custom variable extraction)** | |
| *db_query* | Query the SQLite database for observations with specific codes. |
| *combine_query* | Merge a database query (obtained using db_query) with a cohort, with options to remove observations that do not meet certain criteria, e.g., recorded outside of specified time period, or test value outside of a specified range. |
| *combine_query_boolean* | Combine a database query (obtained using db_query) with a cohort, returning a 0/1 vector depending on whether each individual had a code of interest recorded in a specified time period. |

The process can be done entirely within R using **rcprd** functions, without any specialist understanding of SQLite databases. We recommend this process because once set up, querying the SQLite database is computationally much quicker than reading each of the raw files into the R workspace and querying these separately. It also reduces the probability of errors induced from creating numerous loops through the raw data files. We now move onto a worked example, where we showcase how to implement the above process using **rcprd** and the functions which are detailed in Table 1.

## 3 Worked example for data extraction

### 3.1 Step 1: defining a cohort

We have provided simulated patient, observation and drugissue files which will be utilised in the worked example. The names of the files share the same naming convention given in section 2.1, and column names of the data match the real Aurum data. Numeric variables were simulated at random as integers between 1 and 100, date variables as a date between 01/01/1900 and 01/01/2000, gender as an integer 1 or 2, and year of birth as an integer between 1900 and 2000. Patient ID and practice ID were assigned manually. These files are contained in the *inst/aurum_data* directory of **rcprd**. After installing **rcprd**, either from CRAN (https://CRAN.R-project.org/package=rcprd) or the associated GitHub repository (https://github.com/alexpate30/rcprd), this directory can be accessed using the command system.file("aurum_data", package = "rcprd"). This contains data on 12 fake patients, split across two patient files (*set1* and *set2*) and three observation and drugissue files (all *set1*):

```
#devtools::install_github("alexpate30/rcprd")
#install.packages("rcprd")
library(rcprd)
#> Loading required package: data.table
list.files(system.file("aurum_data", package = "rcprd"), pattern = ".txt")
#> [1] "aurum_allpatid_set1_extract_drugissue_001.txt"
#> [2] "aurum_allpatid_set1_extract_drugissue_002.txt"
#> [3] "aurum_allpatid_set1_extract_drugissue_003.txt"
#> [4] "aurum_allpatid_set1_extract_observation_001.txt"
#> [5] "aurum_allpatid_set1_extract_observation_002.txt"
#> [6] "aurum_allpatid_set1_extract_observation_003.txt"
#> [7] "aurum_allpatid_set1_extract_patient_001.txt"
#> [8] "aurum_allpatid_set2_extract_patient_001.txt"
```

The first step in most analyses is creating and defining a cohort of individuals, which will involve working with the patient files. Data from the patient files can be combined using the extract_cohort function. This will look in the directory specified through the filepath argument, for any file containing "patient" in the file name. All files will be read in and concatenated into a single dataset. In some circumstances, researchers may be provided with a list of patient IDs which meet their inclusion/exclusion criteria. In this case, these can be specified through the patids argument (which requires a character vector). Suppose the individuals meeting the exclusion criteria are those with patid = 1, 3, 4 and 6. We would then specify:

```
pat <- extract_cohort(filepath = system.file("aurum_data", package = "rcprd")
, patids = as.character(c(1,3,4,6)))
str(pat)
#> 'data.frame':    4 obs. of  12 variables:
#>  $ patid        : chr  "1" "3" "4" "6"
#>  $ pracid       : int  49 98 53 54
#>  $ usualgpstaffid: chr  "6" "43" "72" "11"
#>  $ gender       : int  2 1 2 1
#>  $ yob          : int  1984 1930 1915 1914
#>  $ mob          : int  NA NA NA NA
#>  $ emis_ddate   : Date, format: "1976-11-21" "1972-06-01" ...
#>  $ regstartdate : Date, format: "1940-07-24" "1913-07-02" ...
#>  $ patienttypeid : int  58 81 10 85
#>  $ regenddate   : Date, format: "1996-08-25" "1997-04-24" ...
#>  $ acceptable   : int  1 1 0 1
#>  $ cprd_ddate   : Date, format: "1935-03-17" "1912-04-27" ...
```

In other circumstances, a user may need to apply the inclusion and exclusion criteria themselves. In this case, one would initially create a patient file for all individuals.

```
pat <- extract_cohort(filepath = system.file("aurum_data", package = "rcprd")
)
str(pat)
#> 'data.frame':    12 obs. of  12 variables:
#>  $ patid        : chr  "1" "2" "3" "4" ...
#>  $ pracid       : int  49 79 98 53 62 54 49 79 98 53 ...
#>  $ usualgpstaffid: chr  "6" "11" "43" "72" ...
#>  $ gender       : int  2 1 1 2 2 1 2 1 1 2 ...
#>  $ yob          : int  1984 1932 1930 1915 1916 1914 1984 1932 1930 1915
#> ...
#>  $ mob          : int  NA NA NA NA NA NA NA NA NA NA ...
#>  $ emis_ddate   : Date, format: "1976-11-21" "1979-02-14" ...
#>  $ regstartdate : Date, format: "1940-07-24" "1929-02-23" ...
#>  $ patienttypeid : int  58 21 81 10 45 85 58 21 81 10 ...
#>  $ regenddate   : Date, format: "1996-08-25" "1945-03-19" ...
#>  $ acceptable   : int  1 0 1 0 0 1 1 0 1 0 ...
#>  $ cprd_ddate   : Date, format: "1935-03-17" "1932-02-05" ...
```

The cohort of individuals would then be defined by applying study specific inclusion/exclusion criteria. For example, all individuals with > 1 day valid follow up aged 65+, after 1st January 2000. Such criteria can be applied solely using the information available in patient files. In this example, we define the individuals that met the inclusion criteria to be those with patid = 1, 3, 4 and 6.

```r
pat <- subset(pat, patid %in% c(1,3,4,6))
```

The cohort has now been defined. In our hypothetical example we assume there are no inclusion/exclusion criteria dependent on either primary care data or linked data, meaning the answer to Q1 and Q2 are both "no". We therefore move straight to step 5, which is to read in the raw medical/prescription data and create an SQLite database. However, please note, if the answer to Q1 was yes, resulting in the implementation of steps 2 and 3, the process for creating the SQLite database (step 2) will follow the process outlined in section 3.2, and the process of querying this database in order to apply the criteria will follow the process outlined in section 3.3. If the answer to Q2 was yes, and inclusion/exclusion criteria are dependent on linked data, type 1 linked data should be requested from CPRD. This can be read into R and dealt with in the R workspace in order to apply the criteria, without any need for **rcprd** functionality.

### 3.2 Step 5: Reading in data and creating an SQLite database

Data for individuals in the cohort of interest is extracted from the.txt files and put into a SQLite database. This SQLite database is stored on a fixed storage device and can be queried when defining an analysis-ready dataset.

**3.2.1 Add individual files to SQLite database using add_to_database.** The function add_to_database can be used to add individual files to the SQLite database. Start by defining and connecting to your SQLite database. In this article we create a temporary database, but in practice this would be a permanent storage location. Specifically, file.path(tempdir(), "temp.sqlite") would be replaced by the desired file path and SQLite database name.

```r
aurum_extract <- connect_database(file.path(tempdir(), "temp.sqlite"))
```

Next, we add medical diagnoses data from the observation files to this database using the add_to_database function. It is imperative that when adding raw CPRD data to an SQLite database, that the SQLite database itself is stored in a secure environment which aligns with the data storage requirements of CPRD. The simulated raw data provided with **rcprd** can be accessed using the system.file function. The vector of patient id's that defines the cohort is defined through the subset_patids argument. Only data with patid's matching this argument will be added to the SQLite database. The filetype argument will select an appropriate function for reading in the.txt files, and also defines the name of the table in the SQLite database that the files are added to. Note that for the first file, over-write = TRUE is specified to create a new table. For the second and third file, append = TRUE is specified to append to an existing table.

```r
add_to_database(filepath = system.file("aurum_data", "aurum_allpatid_set1_ext
ract_observation_001.txt", package = "rcprd"),
                filetype = "observation", subset_patids = c(1,3,4,6), db = au
rum_extract, overwrite = TRUE)
add_to_database(filepath = system.file("aurum_data", "aurum_allpatid_set1_ext
ract_observation_002.txt", package = "rcprd"),
                filetype = "observation", subset_patids = c(1,3,4,6), db = au
rum_extract, append = TRUE)
add_to_database(filepath = system.file("aurum_data", "aurum_allpatid_set1_ext
ract_observation_003.txt", package = "rcprd"),
                filetype = "observation", subset_patids = c(1,3,4,6), db = au
rum_extract, append = TRUE)
```

We can then query this database using the db_query function and return the first 3 rows. We will showcase how to use this function to query the database for specific codes in section 3.3.3. db_query utilises the package *RSQLite*, and more details on how to query an SQLite database from within R is available in *RSQLite*'s documentation [12].

```
db_query(db_open = aurum_extract, tab = "observation", n = 3)
#>     patid consid pracid  obsid    obsdate   enterdate staffid parentobsid
#>    <char> <char>  <int> <char>     <Date>      <Date>  <char>      <char>
#> 1:      1     33      1    100 1926-05-21 1967-04-13      79          95
#> 2:      1     66      1     46 1932-04-08 1928-04-19      34          17
#> 3:      1     41      1     53 1915-03-29 1994-03-21      35          79
#>           medcodeid value numunitid obstypeid numrangelow numrangehigh
#>              <char> <num>     <int>     <int>       <num>        <num>
#> 1:   498521000006119    48        16        20          28           86
#> 2:         401539014    22         1         2          27            8
#> 3: 13483031000006114    17        78        13          87           41
#>     probobsid
#>        <char>
#> 1:         54
#> 2:         35
#> 3:         74
```

Next, the prescription data from the drugissue files is added to a table called drugissue. A single SQLite database may contain more than one table, so this data is added to a different table within the same SQLite database. The table will take the same name as the filetype argument, unless the table_name argument is specified.

```
add_to_database(filepath = system.file("aurum_data", "aurum_allpatid_set1_ext
ract_drugissue_001.txt", package = "rcprd"),
                filetype = "drugissue", subset_patids = c(1,3,4,6), db = auru
m_extract, overwrite = TRUE)
add_to_database(filepath = system.file("aurum_data", "aurum_allpatid_set1_ext
ract_drugissue_002.txt", package = "rcprd"),
                filetype = "drugissue", subset_patids = c(1,3,4,6), db = auru
m_extract, append = TRUE)
add_to_database(filepath = system.file("aurum_data", "aurum_allpatid_set1_ext
ract_drugissue_003.txt", package = "rcprd"),
                filetype = "drugissue", subset_patids = c(1,3,4,6), db = auru
m_extract, append = TRUE)
```

This table can be queried in the same way, changing the tab argument, which specifies the name of the table in the SQLite database to query:

```
db_query(db_open = aurum_extract, tab = "drugissue", n = 3)
#>     patid issueid pracid probobsid drugrecid  issuedate   enterdate staffid
#>    <char>  <char>  <int>    <char>    <char>     <Date>      <Date>  <char>
#> 1:      1      93      1        88        83 1925-11-15 1967-03-25      98
#> 2:      1      93      1        55        59 1933-07-12 1934-09-07      88
#> 3:      1      16      1        22        82 1946-03-31 1960-04-20      50
#>          prodcodeid dosageid quantity quantunitid duration estnhscost
#>              <char>   <char>    <num>       <int>    <int>      <num>
#> 1: 3092241000033113       58       18          33       27         12
#> 2:   92041000033111       62       93          83       59         11
#> 3:  971241000033111       87       43          83       88         65
```

Listing the tables in the SQLite database shows there are now two, named *observation* and *drugissue*.

```
RSQLite::dbListTables(aurum_extract)
#> [1] "drugissue"   "observation"
```

The add_to_database function allows specification of filetype=c("observation", "drugissue", "referral", "problem", "consultation", "hes_primary","death"), each corresponding to a specific function for reading in the corresponding.txt files with correct formatting. The "hes_primary" options correspond to the primary diagnoses file in linked HES APC data. The "death" file corresponds to the death file in the linked ONS data. If wanting to add other files to the SQLite database, a user defined function for reading in the raw.txt file can be specified through extract_txt_func, and a table name can be specified through tablename. This allows the user to add any.txt file to their SQLite database.

Finally, when manually adding files in this manner, it is good practice to close the connection to the SQLite database once finished.

```
RSQLite::dbDisconnect(aurum_extract)
```

**3.2.2 Add all relevant files to SQLite database using cprd_extract.** In practice, there will be a high number of files to add to the SQLite database and adding each one using add_to_database would be cumbersome. We now repeat the extraction but using the cprd_extract function, which is a wrapper for add_to_database, and will add all the files in a specified directory that contain a string matching the specified file type. Start by creating a connection to a new database:

```
aurum_extract <- connect_database(file.path(tempdir(), "temp.sqlite"))
```

We then use cprd_extract to add all the observation files into the SQLite database. If the connection (aurum_extract) is to an existing database, which is the case here, it will be overwritten when running cprd_extract. The directory containing the files should be specified using filepath. It will only read in and add files with the text string specified in filetype in their file name The filetype argument takes values in c("observation", "drugissue", "referral", "problem", "consultation"). We then query the first three rows of this database, and note they are the same as previously.

```
### Extract data
cprd_extract(db = aurum_extract,
             filepath = system.file("aurum_data", package = "rcprd"),
             filetype = "observation", subset_patids = c(1,3,4,6), use_set =
FALSE)
#>    |
|                                                                      |   0%
[1] "Adding C:/Program Files/R/R-4.4.0/library/rcprd/aurum_data/aurum_allpati
d_set1_extract_observation_001.txt 2024-11-06 09:45:08.286905"
#>    |
|======================                                                |  33%
[1] "Adding C:/Program Files/R/R-4.4.0/library/rcprd/aurum_data/aurum_allpati
d_set1_extract_observation_002.txt 2024-11-06 09:45:08.317863"
#>    |
|===========================================                           |  67%
[1] "Adding C:/Program Files/R/R-4.4.0/library/rcprd/aurum_data/aurum_allpati
d_set1_extract_observation_003.txt 2024-11-06 09:45:08.34949"
#>    |
|======================================================================| 100%

### Query first three rows
```

```
### Query first three rows
db_query(db_open = aurum_extract, tab = "observation", n = 3)
#>      patid consid pracid  obsid    obsdate  enterdate staffid parentobsid
#>     <char> <char> <int> <char>     <Date>     <Date>  <char>      <char>
#> 1:      1     33     1    100 1926-05-21 1967-04-13      79          95
#> 2:      1     66     1     46 1932-04-08 1928-04-19      34          17
#> 3:      1     41     1     53 1915-03-29 1994-03-21      35          79
#>           medcodeid value numunitid obstypeid numrangelow numrangehigh
#>              <char> <num>     <int>     <int>       <num>        <num>
#> 1:   498521000006119    48        16        20          28           86
#> 2:         401539014    22         1         2          27            8
#> 3: 13483031000006114    17        78        13          87           41
#>    probobsid
#>       <char>
#> 1:        54
#> 2:        35
#> 3:        74
```

The process is then repeated for the drugissue files.

```
### Extract data
cprd_extract(db = aurum_extract,
            filepath = system.file("aurum_data", package = "rcprd"),
            filetype = "drugissue", subset_patids = c(1,3,4,6), use_set = FA
LSE)
#>    |
|                                                                  |   0%
[1] "Adding C:/Program Files/R/R-4.4.0/library/rcprd/aurum_data/aurum_allpati
d_set1_extract_drugissue_001.txt 2024-11-06 09:45:08.40585"
#>    |
|======================                                            |  33%
[1] "Adding C:/Program Files/R/R-4.4.0/library/rcprd/aurum_data/aurum_allpati
d_set1_extract_drugissue_002.txt 2024-11-06 09:45:08.433512"
#>    |
|=============================================                     |  67%
[1] "Adding C:/Program Files/R/R-4.4.0/library/rcprd/aurum_data/aurum_allpati
d_set1_extract_drugissue_003.txt 2024-11-06 09:45:08.46284"
#>    |
|==================================================================| 100%
```

```
### Query first three rows
db_query(db_open = aurum_extract, tab = "drugissue", n = 3)
#>      patid issueid pracid probobsid drugrecid   issuedate   enterdate staffid
#>     <char>  <char> <int>     <char>    <char>      <Date>      <Date>  <char>
#> 1:      1      93     1        88        83 1925-11-15 1967-03-25      98
#> 2:      1      93     1        55        59 1933-07-12 1934-09-07      88
#> 3:      1      16     1        22        82 1946-03-31 1960-04-20      50
#>          prodcodeid dosageid quantity quantunitid duration estnhscost
#>              <char>   <char>    <num>       <int>    <int>      <num>
#> 1: 3092241000033113       58       18          33       27         12
#> 2:   92041000033111       62       93          83       59         11
#> 3:   971241000033111       87       43          83       88         65
```

```
### Disconnect
RSQLite::dbDisconnect(aurum_extract)
```

The string to match on, function to read in the raw data, and the name of the table in the SQLite database, can be altered using the str_match, extract_txt_func and tablename arguments respectively. Note the use of str_match may be of particular importance if the naming convention of the raw data differs from what we have described above. The argument rm_duplicates = TRUE can be specified to de-duplicate records before adding into the SQLite database. This will increase computation time, and the derivation of many variables will not be effected by having duplicate records, so consider carefully whether it's necessary to apply this step or not. There will also be the opportunity to de-duplicate the records later on when querying the data. Note that this function may run for a considerable period of time when working with the entire CPRD AURUM database, and therefore it is not recommended to run interactively. While creation of the SQLite database may be time consuming, subsequent queries will be far more efficient, so this is short term pain for a long term gain.

### 3.2.3 Add all relevant files to SQLite database in a computationally efficient manner using the set functionality.

When the number of patients in your cohort is very large (for example millions, or tens of millions), the add_to_database function may perform very slowly. This is because for each observation in the file being added to the SQLite database, add_to_database checks to see whether the patid is contained in the vector subset_patids (a vector of length 20,000,000 in our case). We can utilise the structure of the CPRD AURUM data to speed up this process. If data has the *set* naming convention (see section 2.1), we know that we only need to search for patids from subset_patids, that are in the corresponding patient file. For example, when reading in file *aurum_allpatid_set1_extract_observation_00Y. txt* (for any *Y*), we only need to search whether patid is in the vector of patids from subset.patid, that are also in *aurum_ allpatid_set1_extract_patient_001.txt*, which is much smaller vector. This can reduce the computation time for add_to_ database and cprd_extract.

To achieve this, the subset_patids object should be a data frame with two required columns. The first column should be patid, the second should be set, reporting the corresponding value of set which the patient belongs to. The first step is therefore to create a patient file, which has an extra variable set, the number following the text string *set* in the patient file containing data for that patient. When reading in the patient files to create a cohort, this can be done by specifying set = TRUE. In this example, all individuals in our cohort come from the file with string *set1*, and therefore this variable is the same for all individuals in this cohort, however this will not be the case in practice.

```
pat <- extract_cohort(filepath = system.file("aurum_data", package = "rcprd")
, patids = as.character(c(1,3,4,6)), set = TRUE)
str(pat)
#> 'data.frame':    4 obs. of  13 variables:
#>  $ patid         : chr  "1" "3" "4" "6"
#>  $ pracid        : int  49 98 53 54
#>  $ usualgpstaffid: chr  "6" "43" "72" "11"
#>  $ gender        : int  2 1 2 1
#>  $ yob           : int  1984 1930 1915 1914
#>  $ mob           : int  NA NA NA NA
#>  $ emis_ddate    : Date, format: "1976-11-21" "1972-06-01" ...
#>  $ regstartdate  : Date, format: "1940-07-24" "1913-07-02" ...
#>  $ patienttypeid : int  58 81 10 85
#>  $ regenddate    : Date, format: "1996-08-25" "1997-04-24" ...
#>  $ acceptable    : int  1 1 0 1
#>  $ cprd_ddate    : Date, format: "1935-03-17" "1912-04-27" ...
#>  $ set           : num  1 1 1 1
```

The patient file read in is the same as previously, with the addition of the set column. This file can be reduced to just the patid and set columns, and used as the input to subset_patids when running the add_to_database and cprd_extract functions. When extracting data from observation files with *set1* in the name, it will only search for patient id's with set == 1 in the data.frame provided to subset_patids.

```
### Create connection to SQLite database
aurum_extract <- connect_database(tempfile("temp.sqlite"))

### Add observation files
cprd_extract(db = aurum_extract,
             filepath = system.file("aurum_data", package = "rcprd"),
             filetype = "observation",
             subset_patids = pat,
             use_set = TRUE)
#>    |
|                                                                       |    0%
[1] "Adding C:/Program Files/R/R-4.4.0/library/rcprd/aurum_data/aurum_allpati
d_set1_extract_observation_001.txt 2024-11-06 09:45:08.600991"
#>    |
|=======================                                                |   33%
[1] "Adding C:/Program Files/R/R-4.4.0/library/rcprd/aurum_data/aurum_allpati
d_set1_extract_observation_002.txt 2024-11-06 09:45:08.657204"
#>    |
|===============================================                        |   67%
[1] "Adding C:/Program Files/R/R-4.4.0/library/rcprd/aurum_data/aurum_allpati
d_set1_extract_observation_003.txt 2024-11-06 09:45:08.700486"
#>    |
|=======================================================================| 100%

### Add drugissue files
cprd_extract(db = aurum_extract,
             filepath = system.file("aurum_data", package = "rcprd"),
             filetype = "drugissue",
             subset_patids = pat,
             use_set = TRUE)
#>    |
|                                                                       |    0%
[1] "Adding C:/Program Files/R/R-4.4.0/library/rcprd/aurum_data/aurum_allpati
d_set1_extract_drugissue_001.txt 2024-11-06 09:45:08.726972"
#>    |
|=======================                                                |   33%
[1] "Adding C:/Program Files/R/R-4.4.0/library/rcprd/aurum_data/aurum_allpati
d_set1_extract_drugissue_002.txt 2024-11-06 09:45:08.751347"
#>    |
|===============================================                        |   67%
[1] "Adding C:/Program Files/R/R-4.4.0/library/rcprd/aurum_data/aurum_allpati
d_set1_extract_drugissue_003.txt 2024-11-06 09:45:08.773284"
#>    |
|=======================================================================| 100%
```

The resulting SQLite database is the same as those from sections 3.2.1 and 3.2.2. The computational gains from applying the subsetting in this manner will not be realised in this example given the size of the dataset, however may be worthwhile in practice when working with large cohorts. We do not close the connection, as we will now move onto querying the database to extract variables for creating an analysis-ready dataset.

### 3.3 Step 6: Querying the SQLite database to extract variables

Once the data has been extracted and stored in an SQLite database, it can now be queried to create variables of interest. Please note, if the answer to Q1 was yes, resulting in the implementation of steps 2 and 3, the process for querying the SQLite database to derive variables to apply the exclusion criteria (step 3) will follow the exact same process as outlined here.

The normal process for extracting variables from electronic health records is to create code lists, a group of codes which denote the same condition. The database would then be queried for observations with medical codes matching those in the code list. A variable would then be defined based on this query. Whether this is a binary variable, indicating whether an individual has any record of a given code, or the most recent test result with the given code, or something much more complex. In CPRD Aurum, medical diagnoses and tests are identified from the *observation* file using *medco-deids*, and prescription data is identified from the drugissue file using *prodcodeids*. Creation of code lists is an important step of data extraction, and we refer elsewhere for details on best practice for developing code lists, and the limitations of working with code lists [13–17]. The functions in this section are split into three groups:

• Functions for extracting common variable types.

• Functions for extracting specific variables

• Functions for database queries and custom variable extraction

These functions extract and query the data relative to an *index date*. The index date may be a fixed date (e.g., 1st January 2010), a date which is different for each individual (e.g., date age 50 reached), or a combination of the two (e.g., maximum of 1st January 2010 and date aged 50 reached). Variables are calculated relative to the index date using the observation date (obsdate) in the observation file and the issue date (issuedate) in the drug issue file.

**3.3.1 Functions for extracting common variable types.** There are functions to extract three common variable types, history of condition/medication prior to index date (extract_ho), time from the index date until first occurrence of a medical code/prescription or censoring (extract_time_until), and most recent test result(s) in a given time frame and valid range relative to the index date (extract_test_data).

The first, extract_ho, extracts a binary variable based on whether individual has a specified code recorded prior to index date. This can be applied to search for history of medical diagnoses or prescriptions. The index date must be a variable in the cohort dataset, and is specified through the indexdt argument.

```
### Define codelist
my_codelist_vector <- "187341000000114"

### Add an index date to cohort
pat$fup_start <- as.Date("01/01/2020", format = "%d/%m/%Y")

### Extract a history of type variable using extract_ho
ho <- extract_ho(cohort = pat,
                 codelist_vector = my_codelist_vector,
                 indexdt = "fup_start",
                 db_open = aurum_extract,
                 tab = "observation",
                 return_output = TRUE)
ho
#>    patid ho
#> 1      1  0
#> 3      3  0
#> 4      4  0
#> 6      6  1
```

The second is extract_time_until, which defines a time-to-event/survival variable. This has two components, the time until the first record of a specified code or censoring, and an indicator for whether event was observed or censored. To derive a variable of this type the cohort must also contain a time until censoring variable, which can be specified through censdt.

```
### Add an censoring date to cohort
pat$fup_end <- as.Date("01/01/2024", format = "%d/%m/%Y")

### Extract a time until variable using extract_time_until
time_until <- extract_time_until(cohort = pat,
                                 codelist_vector = my_codelist_vector,
                                 indexdt = "fup_start",
                                 censdt = "fup_end",
                                 db_open = aurum_extract,
                                 tab = "observation",
                                 return_output = TRUE)
time_until
#>   patid var_time var_indicator
#> 1     1     1461             0
#> 2     3     1461             0
#> 3     4     1461             0
#> 4     6     1461             0
```

The third is extract_test, which will extract the most recent test result in a given time frame. The number of days before and after the index date to search for results are specified through time_post and time_prev respectively. Test results are identified from the observation file, using code lists. Lower and upper bounds can also be specified for the extracted data through lower_bound and upper_bound.

```
### Extract test data using extract_test_data
test_data <- extract_test_data(cohort = pat,
                               codelist_vector = my_codelist_vector,
                               indexdt = "fup_start",
                               db_open = aurum_extract,
                               time_post = 0,
                               time_prev = Inf,
                               return_output = TRUE)
test_data
#>   patid value
#> 1     1    NA
#> 2     3    NA
#> 3     4    NA
#> 4     6    28
```

More than one observation can be returned by specifying numobs. Metadata of the test result, such as the unit of measurement, date recorded, and the medical code, can be returned by settings numunitid=TRUE. A variation of this function, extract_test_data_var, will returns the standard deviation of the test data within the specified time and value range. Once all the variables of interest have been extracted, they can be merged into an analysis-ready dataset (step 4).

```
### Recursive merge
analysis.ready.pat <- Reduce(function(df1, df2) merge(df1, df2, by = "patid",
all.x = TRUE), list(pat[,c("patid", "gender", "yob")], ho, time_until, test_d
ata))
analysis.ready.pat
#>   patid gender  yob ho var_time var_indicator value
#> 1     1      2 1984  0     1461             0    NA
#> 2     3      1 1930  0     1461             0    NA
#> 3     4      2 1915  0     1461             0    NA
#> 4     6      1 1914  1     1461             0    28
```

The codelists can also be specified through an R data.frame which must contain either a *medcodeid* or *prodcodeid* column. This may allow the user to run sensitivity analyses more easily if they would like to extract the same variable for different subgroups of the same codelist. For example:

```
my_codelist_df <- data.frame("condition" = "mycondition", medcodeid = c("2215
11000000115", "187341000000114"), "subgroup" = c("subgroup1", "subgroup2"))

extract_test_data(cohort = pat,
                  codelist_df = subset(my_codelist_df, subgroup == "subgroup1
"),
                  indexdt = "fup_start",
                  db_open = aurum_extract,
                  time_post = 0,
                  time_prev = Inf,
                  return_output = TRUE)
#>   patid value    condition     medcodeid    subgroup
#> 1     1    NA         <NA>          <NA>        <NA>
#> 2     3    68  mycondition 221511000000115  subgroup1
#> 3     4    NA         <NA>          <NA>        <NA>
#> 4     6    NA         <NA>          <NA>        <NA>

extract_test_data(cohort = pat,
                  codelist_df = subset(my_codelist_df, subgroup == "subgroup2
"),
                  indexdt = "fup_start",
                  db_open = aurum_extract,
                  time_post = 0,
                  time_prev = Inf,
                  return_output = TRUE)
#>   patid value    condition     medcodeid    subgroup
#> 1     1    NA         <NA>          <NA>        <NA>
#> 2     3    NA         <NA>          <NA>        <NA>
#> 3     4    NA         <NA>          <NA>        <NA>
#> 4     6    28  mycondition 187341000000114  subgroup2
```

Codelists can be specified in this way for any of the functions in this section, or sections 3.3.2 and 3.3.3. However, the extra variables from the codelist (i.e., the condition and subgroup variables in the above example) will only be returned in the output when it is meaningful to do so. For example, in extract_ho, an individual may have many matching codes in their medical history, and therefore it's unclear which should be returned.

### 3.3.2 Functions for extracting specific variables.
There are also a number of functions that can be used to extract specific variables:

- extract_bmi: Derives BMI scores. Requires specification of codelist for BMI, height, and weight separately.

- extract_cholhdl_ratio: Derives total cholesterol/high-density lipoprotein ratio. Requires specification of separate codelists for total cholesterol/high-density lipoprotein ratio, total cholesterol, and high-density lipoproteins separately.

- extract_diabetes: Derives a categorical variable for history of type 1 diabetes, history of type 2 diabetes or no history of diabetes. Requires specification of separate codelists for type 1 and type 2 diabetes. Individuals with codes for both are designated as type 1.

- extract_smoking: Derives a categorical variable for smoking status. Requires specification of seperate codelists for non-smoker, ex-smoker, light smoker, moderate smoker and heavy smoker. If the most recent smoking status is non-smoker, but there are historical codes which indicate smoking, then individual will be classified as an ex-smoker.

It was deemed that these variables required custom functions because their definitions did not fit into any of the variable types from section 3.3.1. In each case, a number of steps are taken in order to clean or manipulate the data in order to get the desired output. For example, height measurements recorded in centimeters are converted to metres in order to calculate BMI scores. This is done through the use of the numunitid variable in the observation file. For both BMI and cholesterol/high-density lipoprotein ratio, the variable can be either be identified directly, or calculated from the component mesaures. In each case, the component parts must be recorded in the specified time range relative to the index date. For smoking status, if an individuals most recent medical observation was recorded as a non-smoker, but their medical record

shows previous smoking, the most recent record is changed to ex-smoker. For diabetes status, diabetes if often recorded with generic codes such as "diabetes mellitus", which does not specify which type. This is dealt with by assuming all generic codes refer to type 2 diabetes, unless that individual also has a specific type 1 diabetes code, in which case they will be determined to have type 1 diabetes as opposed to type 2.

The full details on extracting these variables are provided in the vignette titled Details-on-algorithms-for-extracting-specific-variables. However, it is important to state, that the correct way to define a variable may change from study to study. Therefore when using these functions to extract variables, we encourage taking the time to ensure that the way the variable is extracted matches the definition in ones study, and edit these functions and algorithms accordingly.

### 3.3.3 Functions for database queries and custom variable extraction

These functions are utilised internally in the functions from sections 3.3.1 and 3.3.2. They have been provided to more easily enable package users to write their own functions for extracting variables that are not covered in the previous two sections.

The db_query function will query the SQLite database for observations where the *medcodeid* or *prodcodeid* is in a specified codelist. For example, we can query the *observation* table for all codes with *medcodeid* of 187341000000114. Setting rm_duplicates = TRUE will de-duplicate the output. If the codelist is specified through an R data.frame with the codelist_df argument, the returned query will also contain the variables from the codelist data.frame.

```
my_db_query <- db_query(db_open = aurum_extract,
                        tab ="observation",
                        codelist_vector = "114311000006111")

my_db_query
#>      patid consid pracid  obsid    obsdate   enterdate staffid parentobsid
#>     <char> <char>  <int> <char>     <Date>      <Date>  <char>      <char>
#> 1:       1     41      1    100 1904-09-27  1966-12-02      33          39
#> 2:       3     34      1     79 1976-07-31  1979-03-13      25          90
#> 3:       3     79      1     18 1947-09-30  1927-10-31      93          55
#> 4:       3      7      1     22 1989-07-24  1914-11-30      26          63
#> 5:       4     42      1     43 1924-09-03  1955-03-13      52          93
#> 6:       4     80      1     80 1945-11-26  1993-03-28      48          25
#> 7:       4     43      1     64 1909-09-25  1931-07-17      84          44
#> 8:       6     49      1     96 1932-05-29  1947-10-06      60          86
#> 9:       6     42      1     68 1931-05-20  1978-09-24      81          59
#>           medcodeid value numunitid obstypeid numrangelow numrangehigh prob
#> obsid
#>              <char> <num>     <int>     <int>       <num>        <num>    <
#> char>
#> 1: 114311000006111    18        29        78           4           89
#> 61
#> 2: 114311000006111    83        69        56          75           71
#> 84
#> 3: 114311000006111    85        43         9          61            8
#> 84
#> 4: 114311000006111    16        99        13          64           65
#> 54
#> 5: 114311000006111    59        73        61          89           53
#> 13
#> 6: 114311000006111    88        91        66          20           42
#> 3
#> 7: 114311000006111   100         9        29          71           93
#> 90
#> 8: 114311000006111    15        47        77          26           82
#> 16
#> 9: 114311000006111    81        61        32          67           20
#> 45
```

The combine_query_boolean function will assess whether each individual in a specified cohort (pat) has an observation in the queried data (obtained using db_query) within a specified time frame from the index date, returning a 0/1 vector. The cohort must contain a variable called indexdt containing the index date. This function is useful when defining 'history of' type variables, where we want to know if there is any record of a given condition prior to the index date.

```r
### Add an index date to pat
pat$indexdt <- as.Date("01/01/2020", format = "%d/%m/%Y")

### Combine query with cohort creating a boolean variable denoting 'history of'
combine.query.boolean <- combine_query_boolean(cohort = pat,
                                               db_query = my_db_query,
                                               query_type = "med")

combine.query.boolean
#> [1] 0 1 1 1
```

The combine_query function will merge a cohort with the queried data and return a specified number of observations (numobs) within a specified time frame from the index date. This is useful when extracting test data and requiring access to the values of the tests, or when specifying variables that require > 1 observation within a certain time frame (i.e., two prescriptions within a month prior to index date). For queries from the observation table, the query type can be specified as "med" or "test". Inputting query_type = "med" will just return the date of the observations and the *medcodeid*.

```r
### Combine query with cohort retaining most recent three records
combine.query <- combine_query(cohort = pat,
                               db_query = my_db_query,
                               query_type = "med",
                               numobs = 3)

combine.query
#>      patid        medcodeid     obsdate
#>     <char>          <char>      <Date>
#> 1:       3 114311000006111 1976-07-31
#> 2:       3 114311000006111 1947-09-30
#> 3:       3 114311000006111 1989-07-24
#> 4:       4 114311000006111 1924-09-03
#> 5:       4 114311000006111 1945-11-26
#> 6:       6 114311000006111 1932-05-29
#> 7:       6 114311000006111 1931-05-20
```

For query_type = "test", the value and other relevant information will also be returned, and those with NA values removed (although this can be altered through argument value_na_rm). We then close the connection to the database.

```
### Extract a history of type variable using extract_ho
combine.query <- combine_query(cohort = pat,
                               db_query = my_db_query,
                               query_type = "test",
                               numobs = 3)

combine.query
#>      patid        medcodeid     obsdate value numunitid numrangelow numrangeh
igh
#>     <char>          <char>       <Date> <num>     <int>       <num>        <n
um>
#> 1:      3 114311000006111 1976-07-31    83        69          75
71
#> 2:      3 114311000006111 1947-09-30    85        43          61
8
#> 3:      3 114311000006111 1989-07-24    16        99          64
65
#> 4:      4 114311000006111 1924-09-03    59        73          89
53
#> 5:      4 114311000006111 1945-11-26    88        91          20
42
#> 6:      6 114311000006111 1932-05-29    15        47          26
82
#> 7:      6 114311000006111 1931-05-20    81        61          67
20

### Disconnect
RSQLite::dbDisconnect(aurum_extract)
```

If the query was from the drugissue table, then query_type = "drug" should be specified, and the date of the observations and the *prodcodeid* will be returned. If reduce_output = FALSE, no variables will be removed the output. The functions in this section can be used as building blocks to extract desired variables (e.g., see functions in section 3.3.2).

**3.3.4 Saving extracted variables directly to a disk drive, and utilising rAURUMs suggested directory system.** So far all extracted variables (using functions from section 3.3.1 and 3.3.2) have been read into the R workspace by specifying return_output = TRUE. When working with large cohorts it may be preferable to save the output directly onto a disk drive, by specifying out_save_disk = TRUE. The file path to save the output can be specified manually through the out_filepath argument. However, if this argument is left as NULL, **rcprd** will attempt to save the extracted variable into a directory "data/extraction/" relative to the working directory. The name of the file itself will be dependent on the variable name specified through argument varname. This can be a very convenient way to save the output directly to disk without having to repeatedly specify file paths and file names.

There is similar functionality when specifying the codelists. Codelists can be specified in two ways. The first is to read the codelist into R as a character vector and then specify through the argument codelist_vector, which has been done in all the previous examples. Alternatively, codelists stored on the disk drive can be referred to from the codelist argument in many **rcprd** functions, but requires a specific underlying directory structure. The codelist on the disk drive must be stored in a directory called "codelists/analysis/" relative to the working directory. The codelist must be a.csv file, and contain a column *medcodeid*, *prodcodeid* or *ICD10* depending on the table being queried. The input to argument codelist should just be a character string of the name of the files (excluding the suffix '.csv'). The codelist_vector argument will take precedence over the codelist argument if both are specified.

Finally, there is similar functionality for accessing the SQLite database internally, rather than having to 1) open a connection, 2) use this as an input in the functions, and then 3) remember to close the connection. Instead, if the SQLite

database is stored in a directory "data/sql/" relative to the working directory, the SQLite database can be referred to by name (a character string) with the argument db. A connection to the SQLite datbase will be opened internally within the function call, the SQLite database will be queried, and then the connection closed. Alternatively, a SQLite database stored anywhere on the disk drive can be accessed by specifying the full filepath (character string) with the argument db_filepath.

This workflow is advantageous as it avoids hard file paths which beneficial if wanting to move your code onto another computer system. Furthermore, once codelists and the SQLite database have been created and stored in the appropriate folders, they can simply be referred to by name, resulting in an easier workflow. The function create_directory_system() will create the directory system required to use **rcprd** in this way. To avoid repetition of the previous section, this is show-cased just once using the extract_ho function. For the sake of this example, we start by setting the working directory to a directory called *inst*/*example* within **rcprd**. To maintain the new working directory across multiple R markdown code chunks, we use knitr::opts_knit$set. To follow this section, the user should simply set their working directory as usual using setwd().

Next, the create_directory_system() function can be used to generate the required directory structure within the working directory:

```
create_directory_system()
#> [1] "The working directory is C:/Program Files/R/R-4.4.0/library/rcprd/exa
mple"
```

An SQLite database called "mydb.sqlite" is then created in the "data/sql" directory, using the same data from the previous examples:

```
## Open connection
aurum_extract <- connect_database("data/sql/mydb.sqlite")

## Add data to SQLite database using cprd_extract
cprd_extract(db = aurum_extract,
             filepath = system.file("aurum_data", package = "rcprd"),
             filetype = "observation", use_set = FALSE)
#>   |
|                                                                      |   0%
[1] "Adding C:/Program Files/R/R-4.4.0/library/rcprd/aurum_data/aurum_allpati
d_set1_extract_observation_001.txt 2024-11-06 09:45:09.667303"
#>   |
|======================                                                |  33%
[1] "Adding C:/Program Files/R/R-4.4.0/library/rcprd/aurum_data/aurum_allpati
d_set1_extract_observation_002.txt 2024-11-06 09:45:09.713361"
#>   |
|=============================================                         |  67%
[1] "Adding C:/Program Files/R/R-4.4.0/library/rcprd/aurum_data/aurum_allpati
d_set1_extract_observation_003.txt 2024-11-06 09:45:09.748247"
#>   |
|======================================================================| 100%

## Disconnect
RSQLite::dbDisconnect(aurum_extract)
```

Finally, a code list called *mylist.csv* is created and saved into the *codelists*/*analysis*/ directory.

```
### Define codelist
my_codelist <- data.frame(medcodeid = "187341000000114")

### Save codelist
write.csv(my_codelist, "codelists/analysis/mylist.csv")
```

The *mydb.sqlite* database can now be queried to create a 'history of' type variable using the codelist *mylist.csv*, with the output saved directly onto the disk drive.

```
extract_ho(cohort = pat,
           codelist = "mylist",
           indexdt = "fup_start",
           db = "mydb",
           tab = "observation",
           return_output = FALSE,
           out_save_disk = TRUE)
```

Note that in order to run extract_ho here, a connection to the SQLite database did not need to be created, the codelist did not need to be in the R workspace, and there is no output from this function. Instead the extracted variable has been saved onto the disk drive in an.rds file, and can be read into the R workapce in order to create an analysis-ready dataset using:

```
readRDS("data/extraction/var_ho.rds")
#>   patid ho
#> 1     1  0
#> 3     3  0
#> 4     4  0
#> 6     6  1
```

**3.3.5 Extracting longitudinal data/time varying covariates.** All of the functions in section 3.3.1 and 3.3.2 have the option to extract data at a given time point post index date (specified through the t argument). This allows users to extract data at fixed intervals, which can be utilised for longitudinal analyses where time-varying covariates are required. If saving the extracted variables directly to the disk drive (out_save_disk = TRUE), the time at which data was extracted from, t, will be added to the file name by default.

**3.3.6 Working with linked data.** This worked example has overlooked how to work with linked data. Linked data can be added to the SQLite database. The primary diagnosis file can be added using the add_to_database() function by specifying filetype = "hes_primary". Any other linked file can be added to the SQLite database by writing a user-defined function which reads in the text file and formats the variables appropriately and specifying this through the `extract_txt_ func` argument. However, the user will have to define functions for querying the linked data and creating variables for analysis. Working with linked data has not been made part of the core functionality because it is often much smaller in size, and files (e.g., HES Admitted Patient Care primary diagnosis file, or Office for National Statistics death registration data) are not broken up into large number of smaller files. This means they can be more easily read into R and dealt with in the R workspace.

## 4 Performance and scalability case study

### 4.1 Introduction

So far, all the examples have been run on a very small amount of simulated data which bears no resemblance to the real CPRD Aurum data beyond the structure and variable types (all dates simulated at random between 01/01/1900 and 01/01/2000, all numbers simulated as random integers between 1 and 100). This is due to data access restrictions with the real database, and to allow all examples to be run by package users (a CRAN requirement). In this section we present a case-study to assess the performance of the package in a dataset more representative of the reality of working with large CPRD Aurum data. We set the hypothetical case-study aim of estimating the proportion of individuals alive in 2010 that have a history of hypertension. We compare the performance of **rcprd** with the approach of reading in and querying the raw text files manually in a 'for loop'. Our aim is to quantify elapsed time, peak RAM usage and total RAM usage. We do not compare with **rEHR** or **aurumpipeline**, because we do not expect computational gains over these packages, these packages just do different things.

## 4.2 Data

We utilize the synthetic CPRD Aurum data [18], which is available for free upon submitting a request form. We used the medium-fidelity synthetic CPRD datasets, however, these datasets are not of the same size as the real CPRD data. The synthetic data contains one text file for each file type, the patient file has 45,662 patients, and the observation file contains ~4.5 million observations and is 0.4GB in size. The 2021 CPRD Aurum extract contained > 1200 observation files, each 1GB in size. We therefore had to take steps to increase the size of the synthetic data.

First, we doubled the size of the patient and observation files by duplicating and stacking these datasets, but gave new patient ID's to the new rows. The patient ID's were kept consistent between the two files (i.e., new observations for a given individual in the observation file, were assigned the same patient ID, corresponding to the patient ID of the new individual in the patient file). We then duplicated these files 250 times. This resulted in 250 observation files of size 0.8GB each.

## 4.3 Methods

The code for this case study is available in the supplementary material. With access to the CPRD synthetic data, this example is fully reproducible.

We first created our cohort based off the patient files using the extract_cohort() function. We merged this with the practice file to get the last data collection date for each practice. Start of follow-up was defined using date of registration wit the practice. End of follow-up was the minimum of end of registration with the practice, and practice last data collection. Individuals were excluded if not actively registered on the 1st January 2010. This resulted in ~7 million individuals. When then extracted the variable "history of hypertension" prior to the 1st January 2010 using **rcprd** and a for loop applied to the raw data.

For the **rcprd** approach, we added all the observation files to an SQLite database using cprd_extract(), only adding the data for individuals that met the inclusion criteria, as done in section 3.2.2. This resulted in an SQLite database of size 172GB. We then extracted the history of hypertension variable using the extract_ho() function to query the SQLite database and derive the variable. The for loop approach followed the steps in Box 2.

> ### Box 2:–Steps for estimating a 'history of' variable using the manual for loop approach.
>
> Step 1: Create a 0 vector of length equal to the size of the cohort indicating disease status.
>
> For i in 1:250:
>
> Step 2: Read in raw observation file and format 'obsdate' as a date variable.
>
> Step 3: Subset the observation file to those with medcodeid contained in the codelist (see supplementary material).
>
> Step 4: Merge observation file with the cohort and remove observations that didn't happen prior to the index date.
>
> Step 5: If an individual has a record of hypertension prior to the index date from step 3, set their disease status to '1' in the vector from step 1.
>
> End loop.

We have attempted to do this in the most computationally efficient manner possible. Raw text files are read in using **data.table::fread** which is significantly quicker than **utils::read.table**. Subsetting/filtering is done using **fastmatch::f-match**, which is significantly quicker than using the **base::subset** or **dplyr::filter** functions. In spite of this, we appreciate this approach may still not be optimal. We therefore also apply a third approach, where we just cycle through and read the

250 observation files into the R workspace and subset based on the codelist, and do not apply steps 4 or 5. This acts as a hard lower bound for the computational time when using the manual approach.

Finally, it should be noted that the number of observations in the observation file per patient in the synthetic data is smaller than in the real CPRD Aurum data. This results in a smaller SQLite database for the given cohort, compared to what we would see in practice. Part of this case study is to assess the computational performance of **rcprd** in very large data. We therefore repeated the **rcprd** approach, but when creating the SQLite database, we did not filter to select individuals meeting the inclusion criteria. This resulted in an SQLite database of size 172GB. While we would not recommend this in practice, and therefore performance in comparison to the for-loop approach is not relevant, it gives the user an idea of RAM usage and elapsed time for variable extraction if your SQLite database is much bigger.

## 4.4 Results and discussion

The proportion of individuals with a history of hypertension on 1st January 2010 was 10.34%, obtained either using **rcprd** or the manual approach. Note, this is not representative of the actual prevalence of CPRD on the UK population due to the use of synthetic data. The results comparing computational efficiency in terms of elapsed time and peak RAM usage are contained in Table 2.

The **rcprd** approach took 41 minutes compared to 431 minutes for the manual for-loop approach. The **rcprd** approach also required less than half of the computation time than it took to simply read in the raw text files without any derivation of the variable itself (125 minutes). The **rcprd** approach took 58 minutes to run when querying the SQLite database which did not filter to individuals in the cohort of interest (the SQLite databases were 101GB and 172GB in size respectively).

While the elapsed time of **rcprd** was considerably lower than the for-loop approach, it did have a higher peak RAM (8452 MiB vs 4202 MiB), meaning higher RAM computers may be required to implement this approach. It's worth noting that the peak RAM did not increase when querying the bigger SQLite database. This is because the peak RAM of the **rcprd** approach is driven by the size of the query returned into the R workspace, and subsequent functions to derive the variable, rather than the size of the SQLite database being queried. Querying a larger SQLite database will result in increased computation time, as evidenced by the increase in computation time between the two **rcprd** approaches. However, a relative increase in database size of 1.71 (172GB vs 102GB), only resulted in an increase in computation time of 1.41 (58 vs 41 minutes). We expect a linear relationship between database size and query time given the SQLite query is implemented without indexing.

This is just an example, and results will differ when working with the real CPRD Aurum data. We believe the computational gains in terms of elapsed time will be bigger if running on the entire CPRD Aurum extract, and smaller if running on a smaller cut of the raw data. We expect the elapsed time for the manual method to scale linearly as the size of the raw data increases, as this will result in more text files, and the text files are read in one at a time. On the contrary, we believe the **rcprd** approach will scale linearly with the size of the cohort. We saw previously that this approach scales linearly with the size of the SQLite database, however, observations should only be added to the SQLite database for individuals that meet the cohort inclusion/exclusion criteria. Therefore, the computation time of the functions to query the SQLite database and create variables for analysis (section 3.3) should scale linearly with the size of the cohort. This means if the cohort is very small, and the user is working with the entire raw data, then utilizing the functionality of SQLite becomes particularly relevant.

**Table 2. Elapsed computation time, peak RAM usage, and total RAM usage for each approach.**

|  | rcprd | rcprd (no filtering when creating SQLite) | Manual loop | Manual loop, no derivation of variable |
|---|---|---|---|---|
| Elapsed Computation Time | 41 minutes | 58 minutes | 431 minutes | 125 minutes |
| Peak RAM usage | 8452 | 7370 | 4262 | 3370 |

## 5 Discussion

**rcprd** is an R package which allows users to process CPRD data in R in a consistent and computationally efficiency manner. It provides functionality to both read in and store data, and query this data in order to create analysis-ready datasets. **rcprd** enables the handling and storing the raw data, achieved through the creation of an SQLite database using RSQLite. The user can define their own functions for reading in the raw data, allowing these functions to be applied to other electronic health records, or future versions of CPRD which have different data structures. The functions for extraction of variables to create analysis-ready datasets are split into three groups: 1) Functions for extracting common variable types (history of a specified condition, time until event occurs, or most recent test result); 2) Functions for extracting specific variables; 3) Functions for database queries and custom variable extraction. These querying large data files that could not otherwise be handled in the R workspace. These functions uses computationally efficient SQL queries to query large datasets that could not be read into the R workspace, but no-user knowledge of SQL is required. If using **rcprd** to store data from non-CPRD electronic health records, the functions for variable extraction will not be compatible.

By utilising RSQLite for the storing and querying of the raw data, **rcprd** follows the suggested approach of **rEHR** [7]. In many ways, **rEHR** is more comprehensive than **rcprd**, as it could also be used for case-control matching, cutting up a survival cohort by time-varying covariates, and constructing clinical code lists. Both packages provide functionality to query the underlying database for observations with specific medical or prescription codes without needing SQL experience, however, differ in their method for doing so. **rEHR** functions return observations between specified dates, whether that is all clinical codes, or the first/last clinical code in that period. These functions can also be applied across multiple time periods (i.e., by year) simultaneously. In contrast, **rcprd** functions query the database and return observations in a time period relative to an index date, which may (or may not) be a different date for each patient. As well as functions to query the database, **rcprd** also provides functions which will extract specific variable types, again relative to a given index date. For example, a binary variable based on existence of a clinical code prior to the index date, a test result between a specified upper and lower bound, or a time-to-event/survival type variable. These functions can also be applied any number of days before/after the specified index date to allow extraction of data for longitudinal analyses. The approach of **rcprd**, extracting variables relative to an index date is common when building datasets to be used for development or validation of a clinical prediction model [19], whereas the functions contained in **rEHR** are relevant for a wider range of epidemiological analyses, including case-control studies and reporting descriptive properties such as incidence/prevalence. A final key difference, is that **rEHR** is no longer routinely maintained, and cannot be used out of the box on CPRD Aurum data.

**aurumpipeline** takes a different approach to **rEHR** and **rcprd** by using parquet files to store the data as opposed to SQLite. Parquet files are efficient for data storage and are optimised for query performance, meaning this setup has a high ceiling in term of computational efficiency. **aurumpipeline** provides functions to query the raw data between two fixed dates, with the option to define a binary variable depending if specified medical codes are recorded in this time period. Beyond this, the **arrow** [20] package is recommended for any further data base queries, meaning the derivation of other variables types will require user-developed functions.

The strength of **rcprd** is to simplify the complex process of turning raw CPRD data into an analysis-ready dataset, and does this by following the process of [7]. Functions for extracting variables have been designed to be user friendly, to the extent that all that needs to be specified is the index date and code list, and a number of different common variable types can be derived. More basic functions are also provided, which simply return queries of the underlying data, in order to allow the user flexibility in defining their own functions for extracting other variables or summary statistics. Another limitation is that the scope of this package may not be considered comprehensive and cover the needs of all statisticians/epidemiologists. However, as the scope and size of the package increases, so does the task of maintaining it. We believe in it's current state, maintenance of **rcprd** is manageable going forwards. Furthermore, **rcprd** provides the foundations to build a data set for any type of analysis, some tasks will just require more user-input in order to define new functions around the database queries.

In summary, the main goal of this package is to reduce the duplication of time and effort among those using CPRD data for their research, allowing more time to be focused on other aspects of research projects. **rcprd** will be actively maintained for the foreseeable future. Suggestions for improvement are encouraged and can be posted on GitHub: https://github.com/alexpate30/rcprd.

## Acknowledgments

The authors would like to thank Dr Brian McMillan for his support.

## Author contributions

**Conceptualization:** Alexander Pate, Rosa Parisi, Evangelos Kontopantelis, Matthew Sperrin.

**Formal analysis:** Alexander Pate.

**Funding acquisition:** Matthew Sperrin.

**Methodology:** Alexander Pate, Rosa Parisi, Evangelos Kontopantelis, Matthew Sperrin.

**Software:** Alexander Pate.

**Supervision:** Matthew Sperrin.

**Writing – original draft:** Alexander Pate.

**Writing – review & editing:** Alexander Pate, Rosa Parisi, Evangelos Kontopantelis, Matthew Sperrin.

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
