## [Decision Letter · Decision Letter 0]

12 Mar 2025

Dear Dr. Pate,

Thank you for submitting your manuscript to PLOS ONE. After careful consideration, we feel that it has merit but does not fully meet PLOS ONE’s publication criteria as it currently stands. Therefore, we invite you to submit a revised version of the manuscript that addresses the points raised during the review process.

**Based on the reviewers’ assessments, the manuscript presents a valuable contribution to the field of EHR analysis by introducing the rcprd R package. However, some revisions are necessary before acceptance.**

**Required Revisions:**

**Performance and Scalability – The authors should include a comparison of runtime and RAM usage when extracting data from an SQLite database versus looping through raw .txt files. A practical case study with real CPRD data (if feasible) would further strengthen the manuscript.**

**Comparison with Existing Tools – A direct comparison with other relevant packages, such as rEHR and aurumpipeline, should be included to clarify the advantages of rcprd.**

**Data Quality and Extraction Functions – The authors should clarify how unrealistic date values and general diabetes codes are handled in the extraction functions. Additionally, confirming whether SQLite supports date formats would be useful.**

**Security Considerations – Explicitly state the importance of using a secure device when adding raw CPRD data to an SQLite database.**

We look forward to receiving your revised manuscript.

Kind regards,

B. Sivakumar

Academic Editor

PLOS ONE

**Journal Requirements:**

1. When submitting your revision, we need you to address these additional requirements. Please ensure that your manuscript meets PLOS ONE's style requirements, including those for file naming. The PLOS ONE style templates can be found at https://journals.plos.org/plosone/s/file?id=wjVg/PLOSOne_formatting_sample_main_body.pdf and https://journals.plos.org/plosone/s/file?id=ba62/PLOSOne_formatting_sample_title_authors_affiliations.pdf 2. Please note that PLOS ONE has specific guidelines on code sharing for submissions in which author-generated code underpins the findings in the manuscript. In these cases, all author-generated code must be made available without restrictions upon publication of the work. Please review our guidelines at https://journals.plos.org/plosone/s/materials-and-software-sharing#loc-sharing-code and ensure that your code is shared in a way that follows best practice and facilitates reproducibility and reuse. 3. Thank you for stating the following financial disclosure:  This research was funded by The National Institute for Health Research (NIHR) School for Primary Care Research (SPCR) (reference: NIHR SPCR-2021-2026, grant number 648) and Endeavour Health Charitable Trust. The views expressed are those of the authors and not necessarily those of the NIHR, the Department of Health and Social Care, or Endeavour Health. Please state what role the funders took in the study.  If the funders had no role, please state: "The funders had no role in study design, data collection and analysis, decision to publish, or preparation of the manuscript." If this statement is not correct you must amend it as needed. Please include this amended Role of Funder statement in your cover letter; we will change the online submission form on your behalf. 4. Thank you for stating the following in the Acknowledgments Section of your manuscript: Dr. Brian McMillan was heavily involved with the acquisition of funding which supports the work done in this manuscript. We note that you have provided funding information that is not currently declared in your Funding Statement. However, funding information should not appear in the Acknowledgments section or other areas of your manuscript. We will only publish funding information present in the Funding Statement section of the online submission form. Please remove any funding-related text from the manuscript and let us know how you would like to update your Funding Statement. Currently, your Funding Statement reads as follows: This research was funded by The National Institute for Health Research (NIHR) School for Primary Care Research (SPCR) (reference: NIHR SPCR-2021-2026, grant number 648) and Endeavour Health Charitable Trust. The views expressed are those of the authors and not necessarily those of the NIHR, the Department of Health and Social Care, or Endeavour Health.  Please include your amended statements within your cover letter; we will change the online submission form on your behalf. 5. In the online submission form, you indicated that your data will be submitted to a repository upon acceptance.  We strongly recommend all authors deposit their data before acceptance, as the process can be lengthy and hold up publication timelines. Please note that, though access restrictions are acceptable now, your entire minimal  dataset will need to be made freely accessible if your manuscript is accepted for publication. This policy applies to all data except where public deposition would breach compliance with the protocol approved by your research ethics board. If you are unable to adhere to our open data policy, please kindly revise your statement to explain your reasoning and we will seek the editor's input on an exemption.

Reviewers' comments:

Reviewer's Responses to Questions

**Comments to the Author**

1. Is the manuscript technically sound, and do the data support the conclusions?

Reviewer #1: Yes

Reviewer #2: Yes

2. Has the statistical analysis been performed appropriately and rigorously?

Reviewer #1: Yes

Reviewer #2: N/A

3. Have the authors made all data underlying the findings in their manuscript fully available?

Reviewer #1: Yes

Reviewer #2: Yes

4. Is the manuscript presented in an intelligible fashion and written in standard English?

Reviewer #1: Yes

Reviewer #2: Yes

**Reviewer #1: ** **Summary**

The authors introduce a new R package for simplifying data extraction and cleaning data for CPRD Aurum. Building on older but no longer maintained R packages, the authors present a software package storing the data as SQLite database. This is valuable tool for researchers using large electronic health records, given the time consuming data processing with pre-built functions to extract variables and with some included algorithms for data cleaning.

Overall, the paper and software package are very promising and useful but could be improved by clarifying some of the functionalities of the package and comparing the runtime to existing extraction approaches.

**Comments to the authors**

General comments

All worked examples work well. As a researcher who is regularly using CPRD, the main advantage of using a package as the one presented by the authors would be an improvement of runtime and less local RAM use when cleaning big patient dataset. I was wondering if the authors could present some comparisons of RAM use and runtime by extracting data from the SQLite database versus by looping through the CPRD Aurum raw data .txt files? As someone using datasets with up to 20mio patients, this would be the selling point of using the new R package.

Introduction

As this paper is targeted for an audience which has not used SQLite before, I’d suggest to add a half-sentence to explain how an SQLite database on a fixed storage device can align with the data storage requirements of CPRD usage (e.g., can be stored on a secure sever).

2.2 Recommended process for extraction

Cohort specification is often done with type 1 linkage of the data, i.e., using additional data from HES or the ONS mortality dataset. Could the authors clarify whether it is possible to use linked data for the cohort definition in Step 2.2?

3.2.1 Add individual files to SQLite database using add_to_database

When creating an SQLite database, it would be good to remind the reader that this should be done on a secure device as the following functions add raw CPRD data to the database.

3.2.1 Add individual files to SQLite database using add_to_database

Is it possible to store dates in a date format within the SQLite database? It can be sometimes very helpful to inspect the dataset for unrealistic date values which is more difficult if the dates are stored as a number.

3.3.1 Functions for extracting common variable types

Could the authors clarify some of the functionalities around the extraction functions:

• Is the index date of each function applied to the observation date or the enter date of a medical record?

• Do the function check for unrealistic dates, e.g., events recorded in 1899 or before the patient was born? These are common occurrences in CPRD data and should be removed from the final dataset.

Code lists are often include different categories of codes (e.g., a diabetes code list could contain codes for diagnoses, but also referrals, patient registries, examination results, etc). These different categories are often used in sensitivity analyses but also required if variables are defined based on algorithms, e.g., BMI, vaccinations, eGFR, etc.

I think it would make data extraction easier if the authors allowed the user to enter a data frame for the code list instead of a vector alone. This function could extract based on the medcodeid but would merge the other columns included in the code list data frame to the final output. I know that the same can be done by applying the current functions iteratively but I think it would improve the workflow of data extraction.

Algorithms for data extraction

BMI

I would like to flag that there are entries in CPRD in other units than cm/m and kg/stone (e.g., inches, feet, pounds, …). I would recommend the authors to double check that their algorithm includes all possible measuring units of height and weight used in CPRD.

Diabates status

How does the algorithm handle general diabetes codes, e.g., “diabetes mellitus”, “diabetic foot”, etc? In the current version it only allows to enter specific codes for either type 1 or type 2. However, as the generic codes are commonly used, I would recommend including them.

**Reviewer #2: ** This manuscript introduces rcprd, an R package designed to streamline the extraction, processing, and querying of CPRD data. The package tackles key challenges inherent in working with large-scale EHR data by creating an SQLite database from raw .txt files and offering a suite of functions to extract both common variable types (e.g., history of a condition, time-to-event variables, test results) and specific variables (e.g., BMI, cholesterol/HDL ratio). The manuscript provides a detailed description of each function in the package. The work is timely and promises to significantly reduce redundant efforts in processing CPRD Aurum data, potentially benefiting a wide range of health data researchers. Overall, the manuscript is well-written, well-organized, and represents a significant contribution to the field of EHR analysis.

Major Comments:

Comparison to other packages: Does rEHR or any other package still remain a viable alternative for preparing analysis-ready data from CPRD Aurum data? The authors discussed the differences in approach rather than performance among rEHR, aurumpipeline, and rcprd. Would it be possible to provide a comparison with existing tools? A clear comparison would help highlight the distinct improvements offered by rcprd.

Scalability and practical Implementation: Although the demonstration using a simulated dataset effectively showcases the functionality, the example dataset is relatively small (containing only 12 patients and 8 .txt files). Could the authors expand on practical applications by including performance metrics or a case study using real CPRD data (if feasible) to further validate the package’s utility in real-world settings?

Minor Points:

L165: The authors mention the naming convention for the EHR files. Is user input required for this, or do the CPRD Aurum data files already adhere to the naming convention described in Section 2.1?

L170: Instead of stating “(note, expecting CRAN submission in the next month),” could the authors provide the CRAN link if available? If not, would a specific expected availability date (e.g., year/month) be preferable?

L171: Since reference 12 is the GitHub link for the R package, might it be more straightforward to directly include the GitHub URL rather than citing it as a reference?

L432: When adding all relevant files at once, is there a checking step for duplicate records? Could this potentially pose an issue under certain circumstances?

L581: There appears to be a typo: “ust” -> “must” ?.

**Do you want your identity to be public for this peer review?** For information about this choice, including consent withdrawal, please see our Privacy Policy

Reviewer #1: No

Reviewer #2: No

---

## [Author Response · Author response to Decision Letter 1]

4 Apr 2025

Hello, many thanks to the editor and both reviewers for their comments. We appreciate the time you have given to reviewing this manuscript. We have made considerable changes to the manuscript:

- Added a performance and scalability case study on synthetic CPRD data which is of comparable size to the real CPRD data. This case study is used to assess computational performance and quantify the benefit over the manual ‘for-loop’ approach.

- The ability to specify codelists through R data frames.

- Cleaning tools such as the ability to de-duplicate data, and the removal of dates before 1900 or prior to patient year of birth.

- Dates now returned in date formats after querying database.

- A new flow diagram and process which aligns with the process implemented by CPRD when cohort inclusion/exclusion criteria are dependent on linked data.

- Clarification over how diabetes codes and units of measurements have been handled.

We provide a comment-by-comment response below. In the submitted response document, reviewer/editors comments are in black text, our response is in blue text, and changes to the manuscript in green text. Line numbers refer to the tracked changes manuscript.

Reviewer #1: **Summary**

The authors introduce a new R package for simplifying data extraction and cleaning data for CPRD Aurum. Building on older but no longer maintained R packages, the authors present a software package storing the data as SQLite database. This is valuable tool for researchers using large electronic health records, given the time consuming data processing with pre-built functions to extract variables and with some included algorithms for data cleaning.

Overall, the paper and software package are very promising and useful but could be improved by clarifying some of the functionalities of the package and comparing the runtime to existing extraction approaches.

**Comments to the authors**

General comments

All worked examples work well. As a researcher who is regularly using CPRD, the main advantage of using a package as the one presented by the authors would be an improvement of runtime and less local RAM use when cleaning big patient dataset. I was wondering if the authors could present some comparisons of RAM use and runtime by extracting data from the SQLite database versus by looping through the CPRD Aurum raw data .txt files? As someone using datasets with up to 20mio patients, this would be the selling point of using the new R package.

We have added a comparison of runtime and RAM usage in a practical case study. Please see the new section 4 in the manuscript.

Please note, we have had to delete our raw CPRD Aurum data (common practice once intermediate datasets have been derived for data security reasons), therefore we have made use of the synthetic CPRD datasets for this: https://www.cprd.com/synthetic-data. The synthetic data has the same data structure as the real CPRD Aurum data, however is smaller in size. We have therefore duplicated these synthetic files over and over to create synthetic data which is of comparable size to the real CPRD Aurum data to compare runtime and RAM usage. Another benefit of using the synthetic CPRD data, is that if readers request the data themselves, they will be able to replicate the case study themselves. All code for the case study is available on our GitHub page: XXXX.

Please note, neither the synthetic data or real data can be shared freely, and access is tightly controlled by CPRD. It is therefore not possible to provide easy to access reproducible examples for users that utilise this data and showcase how to use the package. The main body of the paper therefore still utilises the same simulated data from the previous submission.

Finally, while the synthetic data we have created for the case study is still not as big as the data for 20 million patients, it’s on a similar order of magnitude, and the computational time for both methods scales linearly, so we can extrapolate the differences. This is discussed in the final paragraph of section 4. FYI, I work with similarly sized data (~ 20 million patients) and find the SQLite approach to work well. In fact, working with data this size was the motivator for taking up this approach.

Introduction

As this paper is targeted for an audience which has not used SQLite before, I’d suggest to add a half-sentence to explain how an SQLite database on a fixed storage device can align with the data storage requirements of CPRD usage (e.g., can be stored on a secure sever).

You are right about emphasizing data security when utilising rcprd. We have added the following sentence:

Line 82: The SQLite database must be created on a secure device or server which aligns with the data storage requirements of CPRD.

2.2 Recommended process for extraction

Cohort specification is often done with type 1 linkage of the data, i.e., using additional data from HES or the ONS mortality dataset. Could the authors clarify whether it is possible to use linked data for the cohort definition in Step 2.2?

Good question.

Yes, it is possible to use linked data for the cohort definition in step 2.2. Any dataset can be added to the sqlite database using the `add_to_database` function, and specifying a user-defined function (through the `extract_txt_func` argument) for reading in the text file and formatting the variables appropriately. There is also built in function for reading in the primary diagnosis HES file, which can be accessed through specifying `filetype = hes_primary`.

However, we don’t necessarily recommend doing this. The HES data is generally a lot smaller in size and there are little benefits to reading it into an SQLite database and querying this, in comparison to reading in the raw txt file(s) and querying them within R.

Re-reading the proposed steps after your question, we’ve realised it is more appropriate to have a step-by-step process that is in-keeping with the process of requesting of type 1 data, which is very often used for cohort selection. We have re-written the steps and put into a new Box, Box 1. This Box, a Flow diagram, replaces Figure 1.

Line 151: This process aligns with the process implemented by CPRD when cohort inclusion/exclusion criteria are dependent on linked data (see Q2 and step 4).

Finally, please note, because the new steps are contained in a text box, this hasn’t shown up as a tracked change. Not sure why this is the case.

3.2.1 Add individual files to SQLite database using add_to_database

When creating an SQLite database, it would be good to remind the reader that this should be done on a secure device as the following functions add raw CPRD data to the database.

Thanks this is an important suggestion. We have added the following reminder about data security at this point:

Line 268: It is imperative that when adding raw CPRD data to an SQLite database, that the SQLite database itself is stored in a secure environment which aligns with the data storage requirements of CPRD.

Also note, moving forwards, people will have to use the trusted research environment (CPRD Safe), so this will become less of an issue.

3.2.1 Add individual files to SQLite database using add_to_database

Is it possible to store dates in a date format within the SQLite database? It can be sometimes very helpful to inspect the dataset for unrealistic date values which is more difficult if the dates are stored as a number.

The short answer is no, you cannot store variables in date format in the SQLite database. They can be stored either as a number or a character string. When we initially read in the raw text files (using functions in file functions_read_txt.R), we do format them as date values. Upon writing into the SQLite database, it becomes a numeric variable, taking the underlying number that the date value is based on (reference date 1st Jan 1970). When its’ read back in through a query, its read back in as a numeric.

I guess the point at which you would be inspecting the dateset would be after running a database query (i.e. using db_query)? I have changed the code so that db_query will format dates variables as dates. We now also use db_query to view the data, as opposed to RSQLite::dbGetQuery, to ensure that the date formatted values are what is viewed.

Please see the revised section 3.2.1, where all database queries return dates in date format.

Effectively, the date formats are now part of the natural flow of rcprd. The dates are still stored as numbers in the SQLite database, but the user will never come across this unless querying the data using RSQLite, which we don’t recommend or showcase.

3.3.1 Functions for extracting common variable types

Could the authors clarify some of the functionalities around the extraction functions:

• Is the index date of each function applied to the observation date or the enter date of a medical record?

Observation date and issue date. Have added the following text to make this clearer:

Line 666: Variables are calculated relative to the index date using the observation date (obsdate) in the observation file and the issue date (issuedate) in the drug issue file.

• Do the function check for unrealistic dates, e.g., events recorded in 1899 or before the patient was born? These are common occurrences in CPRD data and should be removed from the final dataset.

No they did not. This became very evident in these examples, given the dates are all simulated completely at random.

We have changed the `combine_query` and `combine_query_boolean` functions to remove unrealistic dates (any event before 1900, or before the patient was born). These functions are used internally in all the other functions to extract variables, so will have the desired impact.

Note this has had some minor changes on the results presented.

Code lists are often include different categories of codes (e.g., a diabetes code list could contain codes for diagnoses, but also referrals, patient registries, examination results, etc). These different categories are often used in sensitivity analyses but also required if variables are defined based on algorithms, e.g., BMI, vaccinations, eGFR, etc.

I think it would make data extraction easier if the authors allowed the user to enter a data frame for the code list instead of a vector alone. This function could extract based on the medcodeid but would merge the other columns included in the code list data frame to the final output. I know that the same can be done by applying the current functions iteratively but I think it would improve the workflow of data extraction.

Thanks for this suggestion. I am keen to improve workflow for others, but it’s difficult to predict/know in what ways other people work! We have made the suggested change which has impacted a number of functions in different ways.

The `db_query` function will now merge the output with the codelist dataframe if the codelist is specified using codelist_df. Please note for large queries, this may increase computation time.

For functions extract_time_until and extract_test_data, if the codelist is specified using codelist_df, the output will be merged with the codelist dataframe retaining the other columns from the codelist, as you’ve suggested. For extract_test_data, its the codes associated with the returned tests. For extract_time_until, the associated data is that of the first observation post index date. For all these variables, there will be NA data for individuals who did not have any data matching the codelist.

For all other variables (extract_ho, extract_bmi, extract_cholhdl_ratio, extract_diabetes, extract_impotence, extract_smoking, extract_test_data_var), the codelists can be specified through the codelist_df arguments, but the output will not be merged. This is because these variables are all based off algorithms which may require multiple observations to derive the output, and its unclear which codes to present. E.g. BMI may be derived from directly from a code for BMI, or from two codes of height and weight respectively. Extract_ho might have multiple matching codes in a patients history, again, its unclear which to present.

We have added text in a couple of places, and an example of doing this at the end of section 3.3.1:

Line 776: The codelists can also be specified through an R `data.frame` which must contain either a *medcodeid* or *prodcodeid* column. This may allow the user to run sensitivity analyses more easily if they would like to extract the same variable for different subgroups of the same codelist. For example: ….

Line 849: If the codelist is specified through an R `data.frame` with the `codelist_df` argument, the returned query will also contain the variables from the codelist `data.frame`.

Please let me know if I have misunderstood.

Algorithms for data extraction

BMI

I would like to flag that there are entries in CPRD in other units than cm/m and kg/stone (e.g., inches, feet, pounds, …). I would recommend the authors to double check that their algorithm includes all possible measuring units of height and weight used in CPRD.

You are correct.

We did not include these because we looked at the different unit of measurements for the height and weight records of a large cohort (approx 20 million, real data) of individuals aged 18 - 85 between 2005 – 2020. The results are reported in the article “Details on algorithms for extracting specific variables”. For weight, we found no measurements that were not in kg or stones (or NA). For height, we found no measurements that were not in centimetres or metres (or NA).

We therefore think the approach taken is appropriate.

Diabates status

How does the algorithm handle general diabetes codes, e.g., “diabetes mellitus”, “diabetic foot”, etc? In the current version it only allows to enter specific codes for either type 1 or type 2. However, as the generic codes are commonly used, I would recommend including them.

We assume generic codes refer to type 2, and that type 1 diabetes is only identified through a specific code. This is why if a user has a type 2 and type 1 code, the type 1 code takes precedent, because the type 2 code could just be a generic one.

We have clarified this in the text:

Line 841: For diabetes status, diabetes if often recorded with generic codes such as "diabetes mellitus", which does not specify which type. This is dealt with by assuming all generic codes refer to type 2 diabetes, unless that individual also has a specific type 1 diabetes code, in which case they will be determined to have type 1 diabetes as opposed to type 2.

Full details on the algorithm are available in the package documentation.

Line 846: The full details on extracting these variables are provided in the vignette titled [Details-on-algorithms-for-extracting-specific-variables](https://alexpate30.github.io/rcprd/articles/Details-on-algorithms-for-extracting-specific-variables.html).

We are aware that our algorithms for deriving specific variables are not perfect. There is a lack of convention for how to extract these things, and many researchers may do this in different ways. I was therefore somewhat hesitant to include these in the first place, as it may make the user-overconfident and stop thinking about these things critically themselves. However, we decided to include as it gives the user a number of examples of how to use the functions from section 3.3.3 (db_query, combine_query, etc) to write functions to derive specific variables. We have tried to make it clear that these algorithms should be adapted/changed based on the desired algorithm for defining the variable.

Line 848: “However, it is important to state, that the correct way to define a variable may change from study to study. Therefore when using these functions to extract variables, we encourage taking the time to ensure that the way the variable is extracted matches the definition in one’s study, and edit these functions and algorithms accordingly.”

If you have a specific suggestion about how to deal with generic diabetes codes that is different to how we approach it, please let us know, as we can use it to bolster this argument. We could use it as example of an alternative way to specify the diabetes variable, and then make a broader point about how there are multiple ways to define al

---

## [Editor Report · Decision Letter 1]

12 Jun 2025

rcprd: An R package to simplify the extraction and processing of Clinical Practice Research Datalink (CPRD) data, and create analysis-ready datasets

PONE-D-24-51682R1

Dear Dr. Pate,

We’re pleased to inform you that your manuscript has been judged scientifically suitable for publication and will be formally accepted for publication once it meets all outstanding technical requirements.

Kind regards,

Sreeram V. Ramagopalan

Academic Editor

PLOS ONE